# Generative Adversarial Networks For Data Scarcity Industrial Positron Images With Attention

## Abstract

In the industrial field, the positron annihilation is not affected by complex environment, and the gamma-ray photon penetration is strong, so the nondestructive detection of industrial parts can be realized. Due to the poor image quality caused by gamma-ray photon scattering, attenuation and short sampling time in positron process, we propose the idea of combining deep learning to generate positron images with good quality and clear details by adversarial nets. The structure of the paper is as follows: firstly, we encode to get the hidden vectors of medical CT images based on transfer Learning, and use PCA to extract positron image features. Secondly, we construct a positron image memory based on attention mechanism as a whole input to the adversarial nets which uses medical hidden variables as a query. Finally, we train the whole model jointly and update the input parameters until convergence. Experiments have proved the possibility of generating rare positron images for industrial non-destructive testing using countermeasure networks, and good imaging results have been achieved.

## 1 Introduction

In recent years, with the advancement of science and technology, especially the rapid development of high-end manufacturing, in the field of industrial non-destructive testing, in many cases, it is necessary to perform defect detection without damaging or affecting the performance and internal structure of the device under test. Therefore, there is an increasing demand for corresponding detection devices.

In complex industrial environments (such as aviation, internal combustion engines, chemical engineering, etc.), it is of great research significance to detect faults and defects in closed chambers. In this paper, the use of positron annihilation gamma photon imaging positron emission imaging technology for industrial nondestructive testing is studied. The occurrence of positron annihilation is not affected by factors such as high temperature, high pressure, corrosion, etc., so it can penetrate the dense metal material cavity, realize the undisturbed and non-destructive trace imaging of the detected object, and obtain the detected object after processing. Describe the image and perform a state analysis. Therefore, the quality of imaging technology directly affects the analysis of fault detection results.

Positron Emission Tomography (PET) was first used in medical imaging. The principle is that when a radioactive positron nucleus decays, a proton in the nucleus is converted into a neutron, and a positron and a neutron are released. The positron will quickly combine with the electrons in the material in a very short time, causing a positron annihilation phenomenon, producing a pair of gamma photon pairs with opposite directions and energy of 511KeV. Photon pairs are collected, identified, processed, and finally reconstructed to obtain medical images.

Commonly used PET reconstruction algorithms are analytic method (K, 2000) and statistical method (Shepp & Vardi, 2007). The currently widely used algorithms are MLEM and OSEM. At present, PET technology has been widely used in the clinical diagnosis of human diseases. The advantages are quite obvious, the imaging quality is higher, and it shows great advantages in medical research.

The principle of positron emission in industrial non-destructive fields is similar to medical imaging, but it has its own unique difficulties: the detection environment is more harsh, the sampling time is

short, and at the same time, due to the phenomenon of scattering and attenuation of photons, industrial positron imaging is obtained. The image quality is even worse. Therefore, the reconstructed image needs to be further processed to obtain a higher quality image.

In this paper, we propose adversarial networks of positron image memory module based on attention mechanism. Using medical images as basic data sets, introducing knowledge of migration learning, building memory module according to the contribution of detail features to images, a positron image generation network in the field of industrial non-destructive testing is obtained through joint training, thus achieving higher quality generation of industrial positron images.

In summary, our main contributions in this paper are as follows:

We are the first to advocate an idea of using Generative Adversarial Networks to enhance the detail of the positron image in the industrial non-destructive field, and realize the generation and processing of the scarce image data in the professional field.

We use the medical CT image dataset as the basic training sample of the network framework, which is based on the idea of migration learning, and then extract the features of a small number of industrial non-destructively detected positron images, which can improve the details of the generated images, and make the network model have better applicability in the field of industrial non-destructive testing.

We combine the attention-based mechanism in the professional domain image feature extraction. By constructing a memory module containing industrial positron image features, we can generate image generation in a specific domain, and finally conduct an industrial lossless positron image generation model.

We train the whole network jointly, through the discriminant network of the antagonistic generation network, the front-end network was back-propagated, the input parameters were updated, and the model was optimized. Finally, the convergence was achieved and The Turing test was passed successfully.

## 2 RELATED WORK

Model of GAN: Since GAN (Goodfellow et al., 2014) was proposed in 2014, it has become a research hotspot in the field of unsupervised learning, and related researches continue to increase. For the improvement of network structure, it mainly focuses on training stability and mode collapse. Radford et al.Radford et al. (2015) combine CNN with GAN to realize image generation, which makes GAN move from theory to practice, and establishes the framework and training mode of the whole GAN model. Arjovsky et al. M et al. (2017) propose Wasserstein GAN (WGAN), which use Earth-Mover instead of JS divergence to measure the distance between the real sample and the generated sample distribution, and the problem of pattern collapse was well solved. Mirza proposed Conditional Generative Adversarial Nets (CGAN) in 2014M & S (2014), which transformed unsupervised tasks into supervised tasks, thus improving the stability of training. Mao, X et al.X et al. (2016) propose Least Squares Generative Adversarial Networks (LSGAN), which replace the loss function in the original network with the least squares loss function, alleviating the problem of training instability and insufficient diversity of generated images in GAN. Karras T et al.T et al. (2017) of NVIDIA team propose a progressive structure model to realize the transition from low-resolution to high-resolution image, and the generative model of high definition image can be trained smoothly. DeepMind team (A et al., 2018) introduce the idea of orthogonal regularization into GAN, which greatly improve the generation performance of GAN by truncating the input prior distribution Z in time, and contribute a lot of experience of parameter adjustment in the process of training mode. EBGAN (J et al., 2016) regards the discriminative network in GAN as an energy function and adds reconstruction error to it, which can solve the problem of mode collapse. Unrolled GAN (L et al., 2016) modifies the loss function of the generative network so that the generator can take into account the changes of discriminator after training K times and avoid the mode collapse caused by switching between different modes. DRAGAN (N et al., 2017) introduces the "no regret algorithm" in game theory, and construct gradient penalty scheme to avoid unwanted partial equilibrium. It can solve mode collapse problem and improve training efficiency.

Domain Application: The greatest advantage of GAN is that it can generate real-like samples without any explicit modeling of data distribution in the whole generation process. Therefore, GAN has achieved good results in many fields such as image, text, voice and so on. Image generation(P et al., 2016)(T et al., 2017)(Z et al., 2017), per resolution processing of image(C et al., 2017), object detection(Y et al., 2018)(J et al., 2017), object transfiguration(S et al., 2017)(Wu H, 2017), joint image generation(Liu & Tuzel, 2016), video generation(Chen et al., 2018) (Tulyakov et al., 2017), text to image(H et al., 2017)(A et al., 2017), text generation(J et al., 2018), speech conversion(C et al., 2017), domain adaptation(J et al., 2017), and so on. These researches provide more possibilities for the practical application of GAN in the field.

Medical Image Processing: Image generation in medical field can be divided into two categories: conditional and unconditional. (M et al., 2018)(A & J, 2017)(T et al., 2017) uses DCGAN(Radford et al., 2015) to realize the batch generation of realistic medical images(prostatic lesion plaques, lung cancer cakes, retina), and the resolution reaches 16*16,56*56 and 64*64 respectively, the results of which can pass the Turing test successfully.(C et al., 2018a) uses DCGAN to learn higher resolution MR images distribution from a small number of samples, and compare them with the real images, so as to achieve the effect of "false and true". (C et al., 2018b) uses PGGAN to synthesize skin lesion images, and it successfully showed amazing and highly realistic synthetic images. Conditional medical image generation studies are as follows: (D et al., 2017) realizes CT image synthesis from MR images by means of 3D full convolution network. The loss function of the network is composed of reconstruction loss and gradient loss, and it is also trained by combining additional confrontation network to improve the reality of the synthesized images. (P et al., 2017) compress the images of vascular tree into a multivariate normal distribution by means of the antagonistic automatic encoder. The normal distribution is sampled to synthesize any high resolution vascular tree image, and then the end-to-end framework for high resolution retinal image synthesis is obtained by using image conversion model(P et al., 2016).(T et al., 2017) proposes a two-stage GAN, the first one of which is to synthesize vessel tree images from noise training, and the second network uses image translation framework to generate real, high-resolution pairs of groundtruth vessel segmentation and the corresponding eye fundus image.(W et al., 2018) uses CGAN as the framework to synthesize 200*200 PET images by using CT image and binary label graph. The researchs contribution lies in the construction of a two-channel generation network to achieve a more realistic global output and it called multi-channel GAN. (F & D, 2018) sets up a multi-stage generator to obtain speckle images, low-resolution images, and high-resolution images in turn by intra-vascular ultrasound simulation of tissue maps according to different generating networks. (A & G, 2017) conducts joint learning by adding a specific task network to CGAN, and then obtain a network model that retain specific task characteristics. (M et al., 2018) uses WGAN as the network framework, and use noise and attribute vectors as inputs to generating high-resolution 3D images.

In addition, in the aspects of image segmentation, reconstruction, detection, denoising, recognition, classification, etc, and the use of GAN model has also achieved some research results, which provides feasibility for the application of GAN in professional and specific field. Traditional positron image processing in the field of industrial nondestructive processing focuses on the improvement of image reconstruction algorithm, but due to the inherent characteristics of small sampling points and short time, the quality of the obtained image is poor, and the stability of the image affected by the field is poor. Therefore, in this paper, we innovatively introduce the knowledge of neural network for positron image processing. And through the experimental verification, a high quality image is obtained.

## 3 METHOD

The structure of the whole positron image countermeasure generation network based on memory module is shown in the following figure 1:

### 3.1 STRUCTURE OF GAN

The original GAN(Goodfellow et al., 2014) model consists of two parts: the discriminative nets and the generative nets. Through training, the generator generates pseudo-data as close as possible to the real data distribution. The initial input of the network is random noise signal z, then map it to a new data space by generator to get the generated data G(z), after that the discriminator outputs

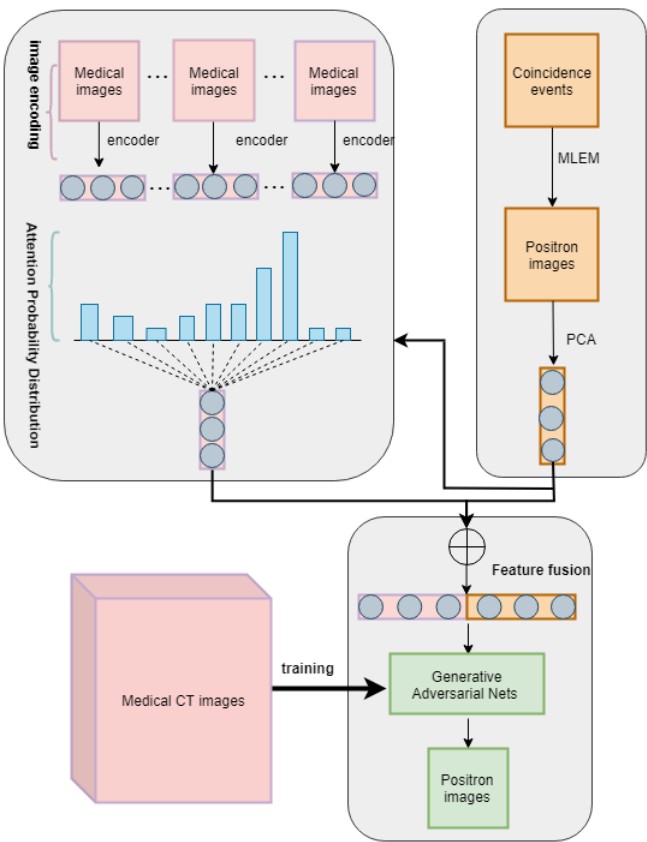

Figure 1: Positron Image Generation Flow Chart.

a probability value through a binary classification of real data and generated data. The confidence level indicates the performance of the generator. Through continuous iteration optimization during the whole training process, the default optimization is achieved when the two sets of data cannot be distinguished.

The basic idea of the GAN comes from the zero-sum game theory. The two networks against each other, in which the discriminator aims to distinguish the real distribution and the generative distribution as far as possible, while the generator aims to make the generated data consistent with the real data in the feature space and make the discriminator indistinguishable, finally the whole model reaches the optimum. The mathematical model is expressed as formula equation 1:

$$\min_{G} \max_{D} V(G, D) = \min_{G} \max_{D} E_{x \sim P_{\text{data}}}[\log D(x)] + E_{z \sim P_x}[\log(1 - D(G(Z)))] \qquad (1)$$

Where $x$ represents real images, $z$ represents noise to the generator, and $G(Z)$ represents the generated images, $D(x)$ represents the probability of judging whether a real image is true.

## 3.2 ENCODER

GAN is used to directly fit the distribution of n-dimensional vectors in images, and sample with random noise, the whole training process is in an unsupervised state, so the direction of data generation is uncontrollable unless the process traverses all initial distributions. At the same time, the generated images may not be the true expression of image meaning due to the excessive pursuit of quality.

Therefore, when using the adversarial model to generate positron images in the industrial non-destructive testing, we choose to add a prior to restrict the data generation, so that the generation model can be trained for specific areas.

Based on the scarcity of industrial positron image data, in this paper, we introduce the knowledge of migration learning and use medical images as training data to construct an encoder, which is based on the variational auto-encoder.

Firstly, we sample medical image data $X$ to get a series of sample points $x_1, x_2, x_3, x_n$, which makes all the sample data in $X$ fit successfully and obtains a distribution $p(x)$, is described as formula equation 2:

$$\mathbb{E} \approx \frac{1}{n} \sum_{i=1}^{n} x_i \quad x_i \sim p(x) \tag{2}$$

In order to achieve this goal, the distribution fitting of data sample $X$ is finally realized with the help of implicit variable $Z$. It is assumed that $p(x)$ describes a probability distribution of $X$ generated by $Z$, which satisfies the Gauss distribution. Therefore, the whole encoder can be expressed as sampling $Z$ from the standard normal distribution. In the process, we can get Variance and Mean of Sample Data $X$ Sampled by Neural Network. The clustering process can be parametrized as formula equation 3:

$$\begin{aligned} \mu_k &= f_1\left(X_k\right) \\ \log \sigma^2 &= f_2\left(X_k\right) \\ p(Z) = \sum_X p(Z|X)p(X) &= \sum_X \mathcal{N}(0,1)p(X) = \mathcal{N}(0,I) \end{aligned} \tag{3}$$

Where the mean and variance of the normal distribution which is exclusive to $X_k$ can be obtained. Then $Z_k$ can be sampled from this exclusive distribution.

### 3.3 FEATURE EXTRACTION-MEMORY MODULE

In order to obtain a more suitable generation model for positron image in the industrial field, we proposes an image feature memory module based on attention mechanism, which is used to extract domain image features. This network is a research focus of this chapter.

The basic flow of the whole network is as follows:1) use neural network to extract the feature of the rare positron images, and obtain the images feature vectors. 2) combine the vector and the hidden variable get in Section 3.2 based on attention mechanism to obtain an image memory model. 3)use the memory model as the input of adversarial nets and train jointly with the whole network to obtain positron image generator for the industrial field.

Positron image feature extraction We use the principal component analysis(H et al., 2015) is used to extract the positron sample data and the vector space transformation is used to reduce the dimensionality of higher dimensional positron data. Firstly, transform the original data into a new coordinate system by projection according to the new coordinate vector. Secondly, the variance of the first principal component of the projection data in the new coordinate system is the largest. As the dimension increases, the variance decreases in turn and the dimension decreases. is described as formula equation 4:

$$Y = \begin{bmatrix} y_1^T \\ y_2^T \\ \cdots \\ y_m^T \end{bmatrix} = \begin{bmatrix} y_{1,1} & y_{1,2} & \cdots & y_{1,n} \\ y_{2,1} & y_{2,2} & \cdots & y_{2,n} \\ \vdots & & & \\ \vdots \cdots & \cdots & \cdots & \cdots \\ y_{m,1} & y_{m,2} & \cdots & y_{m,n} \end{bmatrix}_{m \times n} \tag{4}$$

$m$ represents the positron sample, $n$ represents the sample dimension and the sample $Y = m \times n$ .

The data matrix $Y$ is de-averaged so that the mean value of each dimension is 0. Then we should find the most important feature vectors in the images, that is, the data on the coordinate axis represented by the feature fluctuates the most and the sum of squares of all samples projected on unit vector $\mu$ is

the largest. Then we get the value of $\mu$ using Lagrange theorem. The mathematical expression is as follows equation 5:

$$
\begin{aligned}
u^* = \arg\max \frac{1}{m} \sum_{i=1}^m \left(y_i^T u\right)^2 &= \operatorname{argmax} u^T \left(\frac{1}{m} \sum_{i=1}^m y_i y_i^T\right) u^N \\
\mathrm{L}(\lambda, u) &= u^T \Sigma u + \lambda \left(u^T u - 1\right) \\
\frac{\partial L}{\partial u} &= \Sigma u - \lambda u^j \\
u^* = \arg\max u^T \Sigma u &= \operatorname{argmax} \lambda u^T u
\end{aligned}
\tag{5}
$$

The nets use convolution neural network(CNN) to construct image feature extraction network, the network structure is divided into three layers, namely two convolution layers and one non-linear output layer. Firstly, small image slices are extracted from sample images and the dimension of the slices is the same as convolution core. Then traverse all the pixels in them and perform two-level convolution operation. Finally, hashing operation and histogram statistics are carried out in the output layer to print the feature vector.

Memory Module Based on Attention Mechanism The obtained positron eigenvector $\hat{Y}$ is fused with the hidden variable $Z$ of the medical image obtained based on the attention mechanism to get the input nets. The purpose is to make the prior knowledge contained in the nets more focused on positron features, so that the features of scarce data can be more applied in the whole training process.

The basic idea is the global attention and the focus in our model is to extract all positron image features. The specific realization is to align image data vectors, directly use medical images as query vectors, and input positron image feature vectors as hidden state to calculate their weights equation 6.

$$
\begin{aligned}
a_t(s) = \operatorname{alig} n\left(z_t, \overline{y}_s\right) &= \frac{\exp(s\,\mathrm{core}(z_t, \overline{y}_s))}{\sum_{s'} \exp(s\,\mathrm{core}(z_t, \overline{y}_{st}))} \\
\mathrm{score}\left(z_t, \overline{y}_s\right) &= z_t^T W_a \overline{y}_s
\end{aligned}
\tag{6}
$$

Where $z_t$ is the medical image distribution, $\overline{y}_s$ are feature vectors extracted from positron images, and $\mathrm{score}\left(z_t, \overline{y}_s\right)$ is the scoring criterion for the operation.

We get a constant and normalize it. And the contribution degree of each feature of positron image to the network can be obtained. So the image feature can be fused according to the weight ratio. Finally, the vector containing prior knowledge in the field is obtained as overall input of adversarial nets.

## 3.4 ADVERSARIAL NETS

Generative model The generative network is constructed based on DenseNet (G et al., 2017), and the positron image features can be requisitioned repeatedly in the model. The network can also strengthen the contribution of the characteristics of scarce data so that the generated images closer to real industrial positron images in detail.

The generative model is as follows: the output of the memory model in chapter 3.3 as a whole input to the net, and the input of each layer is related to the output of all the previous layers, not only related to upper layer. It can be expressed as formula equation 7:

$$
X_l = H_l\left([X_0, X_1, \cdots, X_{l-1}]\right)
\tag{7}
$$

$\left[X_0, X_1, \cdots, X_{(l-1)}\right]$ is the concatenation to the net. We can group all output feature maps from layer $X_0$ to $X_{(l-1)}$ according to different channels and the structure is used to reduce the parameters without losing features randomly, so that the initial input can enter each layers convolution calculation to realize the feature reuse. The basic structure is 33 convolution layer, Batch Normalization(S & C, 2015)and ReLU non-linear activation layer.

Feature maps of all previous layers need to be cat in the network. In order to perform the down sampling operation, the net is divided into several Denseblocks and transition layers are used between different them. Referring to the original network, it consists of Batch Normalization layer,

1*1 convolution network and 2*2 average-pooling. In the same Denseblock, the state of each layer is associated with all previous layers, and the training of each layer is aimed at the global state feedback of the network to update the parameters.

Discriminative model The discriminative net is used to discriminate specific images in a specific domain, in which domain image features can be used as the evaluation criteria for network classification as much as possible. The net uses Markov Model based on PatchGAN, which is composed of full convolution layers. The output is an n-dimensional matrix, and the mean of the matrix is used as the output of the discriminative network, so that each receptive field in the image can be judged, which is equivalent to the convolution discriminant in batches by layers, and finally fed back to the whole network. The calculation of receptive field is equation 8:

$$size_{input} = (size_{output} - 1) * k_{stride} + k_{size} \tag{8}$$

In the model, the real input samples are medical data sets. Therefore, in order to make the generated data better characterize positron image features, we need to add an additional attention perception loss function to the net. The loss function of the whole net consists of two parts: $L_G AN$ and $L_A PG$ , the attention loss function $L_A PG$ is used to characterize the feature contribution of positron image, which is realized to measure the distribution distance between the generated data and the positron images. The loss function is described as equation 9:

$$L_{APG} = E_{x,a \sim p(x,a)} \sum_{i=1}^{s} \frac{1}{W_i} \|D'(x) - D(G(a))\| \tag{9}$$

$W_i$ represents number of elements in each layer, and s is the number of layers. The loss function of the whole net can be described as equation 10, and the $L_G AN$ is similar as the original GAN.

$$L_{GAN*} = L_{GAN} + L_{APG} \tag{10}$$

## 4 EXPERIMENTAL RESULTS

### 4.1 IMPLEMENTATION DETAILS

We design the model firstly by using encoder to obtain the medical images hidden vectors and using principal component analysis to reduce positron images dimensionality and extract the main feature. Train memory module and adversarial nets jointly, and in the process of backpropagation, the identification network updates the parameters of the front-end network, so that the feature extraction network extracts the features repeatedly until the whole network achieves the optimal model. Finally, the positron image generator for industrial non-destructive testing is obtained.

The discriminator refers to pixel and each batch is 70*70.The learning rate is 0.0002 in the whole net. The model are trained iteratively using Adam algorithm(=0.5).

### 4.2 DATASET

DeepLession The dataset(K et al., 2018) contains more than 32,000 lesions from more than 10,000 case studies. The data is developed by National Institutes of Health Clinical Center(NIHCC), and developed by mining historical medical data from image archives and communication systems. It is now maybe the largest set of CT medical image data available to all. In the experiment, 150,000 images were selected for our training and the image pixels are 256*256.

Positron images The dataset is obtained by the Geant4 Application for Tomographic Emission(GATE). GATE is a simulation software which can simulate the physical process of PET imaging system, such as geometric size, material composition, detector movement and so on. In the model design, the anisotropic tube made of aluminium metal is filled with positron nuclide solution. Its activity is 600 Bq, the number of detectors is 184 *64, the sampling time is 0.1 s, the energy resolution is 15 percent, the time resolution is 300 ps, the energy window is 350-650 keV, and the time window is 10 ns. The design sampling time is 0.1s to meet the needs of rapid sampling in

Table 1: Evaluation results

| METHOD | MS-SSIM | FID |
|--------|---------|-----|
| VAE | 0.0485 | |
| DCGAN | 0.0522 | 15.2 |
| WGAN | 0.0567 | 27.5 |
| Ours | 0.0694 | 52.8 |

the industrial field. Using Maximum Likelihood - Expectation Maximization (MLEM) iteration algorithm to realize image preliminary reconstruction and obtain positron defect image in industrial field.

### 4.3 EVALUATION

In this section, we compare our approach with the commonly used generation model, aiming at the generation of industrial positron images. Quantitative results are presented in Table 1.

The metrics to evaluate the performance of our method, namely multi-scale structural similarity(MS-SSIM) (A et al., 2017) and Frchet Inception Distance (FID)(M et al., 2017). The SSIM is used to measure the similarity of two images and FID is to measure the Frchet distance of two distributions in feature space.

By comparing the experimental data, we can clearly see that the confrontation network constructed in this paper has a better effect on the generation of positron images for professional fields, and the generated images are closer to the real images.

## 5 CONCLUSION

In this paper, we introduce an application of GAN in the field of nondestructive testing for specific industries. We combine the knowledge of transfer learning to make up the problem of insufficient data. The key point is to introduce attention mechanism to construct a positron image feature memory module, which can reuse image features under the condition of scarce data. At the same time, an attention loss function is added to the discriminative net to further improve the generator performance. Experiments show that compared with the start-of-the-art generation methods in deep learning, the model in our paper has an obvious improvement in the quality of industrial positron image generation.

In the future, our focus is to further study the application of generative adversarial networks in industrial positron image processing, and to further improve the quality of domain images.

### ACKNOWLEDGMENTS

This research has been funded by the Natural Science Foundation of China (No.51875289No.61873124), the Aeronautical Science Foundation of China (No.2016ZD52036), the Fundamental Research Funds for the Central Universities (No. NS2019017).

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

## A    APPENDIX

The Positron image are shown in figure 2, which are simulation diagram with defects. The picture shows one-to-one correspondence between the original images and the generated images.

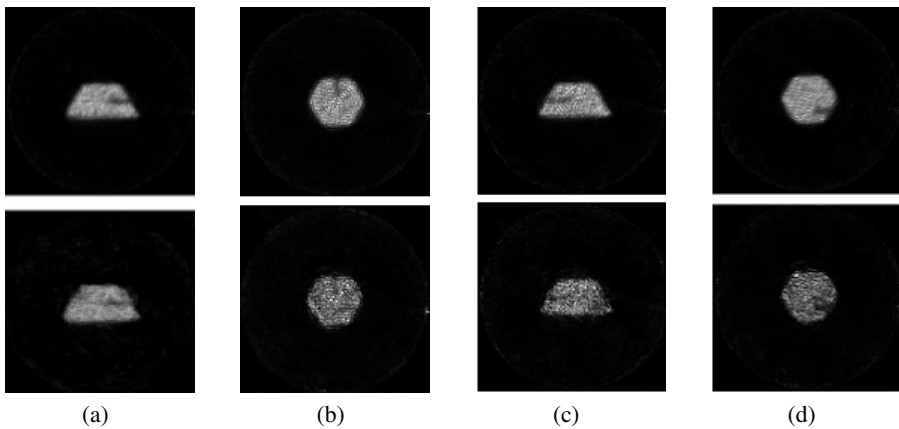

(a)       (b)       (c)       (d)

Figure 2: positron images

