# OpenReview forum: "Generative Adversarial Networks For Data Scarcity Industrial Positron Images With Attention"
_ICLR.cc/2020/Conference — Reject_

### Official Review · AnonReviewer3 · 2019-10-16
**Official Blind Review #3**

**Rating:** 1

**Review:**

The topic of the paper is a GAN framework to enhance PET images in industrial inspection, as far as I understand by transfer learning from a medical PET database. Unfortunately, I am unable assess the paper due to serious language problems. The text is incoherent and not understandable, it is impossible to decipher what is actually proposed.

Additional Comments:

The text reads like a machine translation gone wrong, including weird terms like "migration learning" (transfer learning?), "antagonistic generation network" / "countermeasure generation network" / "confrontation network" (GAN?).

References are also mangled and undecipherable. And seemingly also not always well-chosen - even if I cannot map it to a paper due to bibtex problems, it appears implausible to reference a 2015 paper for something as basic as principal component analysis.

It seems that the only experiments are on simulated PET data. In may view that would be insufficient for a largely empirical application paper.

The paper claims to be quantitatively better than the baselines, but has, by far, the highest Frechet Inception distance. To my understanding, FID is a distance, lower is better.



**Experience Assessment:**

I have published one or two papers in this area.

**Review Assessment: Checking Correctness Of Derivations And Theory:**

N/A

**Review Assessment: Checking Correctness Of Experiments:**

I assessed the sensibility of the experiments.

**Review Assessment: Thoroughness In Paper Reading:**

I read the paper at least twice and used my best judgement in assessing the paper.

---

### Official Review · AnonReviewer1 · 2019-10-23
**Official Blind Review #1**

**Rating:** 1

**Review:**

This work proposes a method to transfer information from PET imaging data from the medical domain, where data is highly available, to the industrial one, where data is scarce, in the context of non-destructive material quality evaluation. The basic idea seems interesting, but unfortunately in the present form he paper is very difficult to appreciate, as it lacks of important details concerning methodology, experimental results, and comparison with respect to the state-of-the-art. Moreover, the manuscript still appears in a draft form. Sentences are often broken, the text presents many typos and grammar mistakes, and the citations are not understandable.

Missing methodological details are grouped in the following parts of this review.

Section 3.2 Encoder
Paragraph 2: The authors claim that they introduce the knowledge of migration learning without explaining it. Where does this concept come from? Is there a literature about it or is this a new concept?

Medical images are fitted though a variational auto encoder (VAE) (citation missing). The encoder description is minimal and lacks of implementation details (see Eq. 3), while the decoder description is missing throughout the paper.

Eq. (2): the authors write that they obtain a distribution p(x) according to Eq. (2), but this equation is just the formula for the sample mean, where p(x) is sampled and not computed.

Eq. (3): f1 and f2 are never made explicit in the paper so we do not know if they are linear or non-linear functions. The prior p(Z) is decomposed as a summation of posteriors p(Z|X) and the choice to have these posteriors equal to N(0,1) (1-dimensional, which is unusual) regardless of the data point X is not explained.

Section 3.3 Feature extraction memory module
Feature extraction from industrial images is done through principal component analysis (PCA). In the same paragraph is written that features are extracted through convolution neural networks (CNN). So it is not clear if there is an arbitrary choice to use PCA or CNN, and what are the conditions when this happens.

Eq. (6): the score between z_t (medical image distribution) and y_s (industrial image feature vector) is computed as scalar product dot(z_t , W_a * y_s). The key link is the linear mapping W_a, which is never made explicit in the paper. How do the authors compute W_a ?

Section 4.1 Implementation details
the networks called “identification network” and “front-end network” are not well defined. They may refer to the  VAE, the CNN, the Adversarial Nets. There is too much ambiguity. A captioned figure can help in resolving the ambiguities.

Section 4.2 Dataset
In the first paragraph the authors cite a dataset of CT images, while the main focus of the paper is on PET images. How the CT images comes into play?

Section 4.3 Evaluation
The authors compare their method with respect to other three methods, namely VAE, DCGAN, and WGAN.
The implementation details of the competing methods are not described so we cannot be sure about the fairness of the comparison.


Other considerations
Introduction, 4th paragraph: “imaging quality is higher” With respect to what? Usually PET images have the worst quality in the medical domain.
Introduction, 3rd-to-last paragraph: “We use the medical CT image ...”. Should be PET images

Related work: This section is a list of works and the relation with the current paper is not highlighted.
“lung cancer cakes” what are they?

Paragraph 3.2: “excessive pursuit of quality” why is it bad? “migration learning” do they mean transfer learning? Equation 2 and 3 in relation to a clustering problem never pointed out before in the paper.

Paragraph 4.1. What is the meaning of “Adam algorithm(=0.5)” ?

Figures 1 and 2 have very minimal captions. What do they represent?

Citation formatting problem: name and surname are switched. Journal/conferences often omitted.

**Experience Assessment:**

I have published in this field for several years.

**Review Assessment: Checking Correctness Of Derivations And Theory:**

I carefully checked the derivations and theory.

**Review Assessment: Checking Correctness Of Experiments:**

I carefully checked the experiments.

**Review Assessment: Thoroughness In Paper Reading:**

I read the paper thoroughly.

---

### Official Review · AnonReviewer2 · 2019-10-24
**Official Blind Review #2**

**Rating:** 1

**Review:**

This paper applies GAN to the field of nondestructive testing for specific industries. This paper is more like a technical report rather than a formal paper. It seems to me that this paper should be submitted to other computer vision conferences of even specific areas while not in ICLR.

Issues:

* Bad format or organization. The authors are suggested that they should give a subtitle of each categories of work in the related work section. The table and picture in this paper are arbitrarily designed and take too much space. In some equations, you need to put commas to separate different equations.

* Acknowledgements reveal personal information in the paper. It's not allowed in ICLR submission that would review the identity of the authors. This is a highly critical issue.

* Bad writing. I can barely understand what the authors are talking about. For example, "In this paper, we propose adversarial networks of positron image memory module based on attention mechanism". Are you referring you proposed a new GAN model in the paper? Or you proposed a positron image memory module? ...

* What are the functions of the metrics used in the experimental part? The higher, the better? Or the lower, the better? Besides, you need some analysis to illustrate the significance of your results.

Considering these issues demonstrated in the paper, I recommend rejection.

**Experience Assessment:**

I do not know much about this area.

**Review Assessment: Checking Correctness Of Derivations And Theory:**

I did not assess the derivations or theory.

**Review Assessment: Checking Correctness Of Experiments:**

N/A

**Review Assessment: Thoroughness In Paper Reading:**

I read the paper at least twice and used my best judgement in assessing the paper.

---

### Decision · Program_Chairs · 2019-12-19

**Decision:**

Reject

**Comment:**

The paper studies Positron Emission Tomography (PET) in medical imaging. The paper focuses on the challenges created by gamma-ray photon scattering, that results in poor image quality. To tackle this problem and enhance the image quality, the paper suggests using generative adversarial networks. Unfortunately due to poor writing and severe language issues, none of the three reviewers were able to properly assess the paper [see the reviews for multiple examples of this]. In addition, in places, some important implementation details were missing.

The authors chose not to response to reviewers' concerns. In its current form, the submission cannot be well understood by people interested in reading the paper, so it needs to be improved and resubmitted.